# The TGF-β/UCHL5/Smad2 Axis Contributes to the Pathogenesis of Placenta Accreta

**DOI:** 10.3390/ijms241813706

**Published:** 2023-09-05

**Authors:** Kei Hashimoto, Yuko Miyagawa, Saya Watanabe, Kazuki Takasaki, Miki Nishizawa, Keita Yatsuki, Yuko Takahashi, Hideo Kamata, Chikara Kihira, Haruko Hiraike, Yukifumi Sasamori, Koichiro Kido, Eiji Ryo, Kazunori Nagasaka

**Affiliations:** Department of Obstetrics and Gynecology, Teikyo University School of Medicine, Tokyo 173-8605, Japan

**Keywords:** angiogenesis, cell invasion, extracellular signal-regulated kinase, placenta accreta spectrum, placentogenesis trophoblast, Smad

## Abstract

Placenta accreta is a high-risk condition causing obstetric crisis and hemorrhage; however, its pathogenesis remains unknown. We aimed to identify the factors contributing to trophoblast invasiveness and angiogenic potential, which in turn drive the pathogenesis of placenta accreta. We focused on the transforming growth factor (TGF)-β1-Smad pathway and investigated the intrinsic relationship between the time- and dose-dependent inhibition of the ubiquitinating enzyme UCHL5 using bAP15, a deubiquitinase inhibitor, after TGF-β1 stimulation and the invasive and angiogenic potential of two cell lines, gestational choriocarcinoma cell line JEG-3 and trophoblast cell line HTR-8/SVneo. UCHL5 inhibition negatively regulated TGF-β1-induced Smad2 activation, decreasing extravillous trophoblast invasiveness. Smad1/5/9 and extracellular signal-regulated kinase (ERK) were simultaneously activated, and vascular endothelial growth factor was secreted into the trophoblast medium. However, extravillous trophoblast culture supernatant severely impaired the vasculogenic potential of human umbilical vein endothelial cells. These results suggest that the downstream ERK pathway and Smad1/5/9 potentially regulate the TGF-β1-Smad pathway in extravillous trophoblasts, whereas Smad2 contributes to their invasiveness. The abnormal invasive and angiogenic capacities of extravillous cells, likely driven by the interaction between TGF-β1-Smad and ERK pathways, underlie the pathogenesis of placenta accreta.

## 1. Introduction

Placenta accreta was first described by Irving et al. in 1937 as a condition in which the placenta fails to detach during delivery [1]. In recent guidelines, the terms “placenta accreta”, “increta,” and “percreta” are used to describe the entire placenta accreta spectrum (PAS) [2]. PAS is thought to be caused by the inability to suppress chorionic invasion due to the lack of formation of a placental abruption membrane on the placental attachment surface or the inability of the abruption membrane to develop because of scar tissue on the uterine wall, resulting in a placenta that is firmly attached to the uterine muscle and cannot be detached [3]. PAS is often associated with pregnancy complications, such as placenta previa, which in turn may lead to obstetric hemorrhage, hemorrhagic shock, and disseminated intravascular coagulation, and is also a cause of maternal death. However, recent reports have indicated that half of women with PAS do not have placenta previa or a history of cesarean delivery and may have a good prognosis and successful outcome [4]. Compared to other placenta-related disorders of pregnancy, such as preeclampsia or fetal growth restriction, few studies have investigated the etiopathogenesis of PAS. Molecular studies on placentation have been conducted both in vitro and in vivo, focusing on the pathological mechanisms of pregnancy-induced hypertension and preeclampsia [5,6,7]. Both these pregnancy complications can cause severe perinatal complications that require life-saving treatment for the mother and infant [8].

Abnormalities during placental development in the first trimester of pregnancy cause both PAS and preeclampsia [9] and are highly associated with gestational hypertension-preeclampsia [10]. Vascular remodeling is deficient in PAS because endothelial cells induced by extravillous trophoblasts are prevented from penetrating the maternal adventitia and myometrium. Extravillous trophoblasts, and their epithelial–mesenchymal transition (EMT), are key in uterine spiral arteriole remodeling during pregnancy, an event that is critical for successful pregnancy outcomes. Some histopathological studies have shown that EMT-associated proteins such as E-cadherin, vimentin, Snail, and transforming growth factor-β (TGF-β) are highly expressed in the villi and desmoplasia in placenta accreta tissues [11,12]. However, molecular studies on the mechanism of extravillous trophoblast invasion in placenta accreta and the dysplasia of vascular remodeling have not identified the mechanisms underlying the pathogenesis of PAS. The mechanisms contributing to PAS and carcinogenesis are thought to be similar in terms of pathological aspects such as invasiveness, proliferation, and angiogenesis [3], suggesting that abnormalities in the invasive and angiogenic potential in the placenta may occur in a similar fashion; however, no underlying mechanisms have been elucidated.

Previous studies have shown that TGF-β1 plays an important role in extravillous trophoblast invasion, proliferation, differentiation, and uterine spiral artery remodeling [11,13,14]. Interestingly, some studies have reported associations between TGF-β1 polymorphism, plasma levels, and the risk of preeclampsia [15]. No definitive conclusion has yet been drawn, and studies on TGF-β1 levels in preeclampsia have reported contrasting results [7,16,17,18,19]. Based on these previous studies, we hypothesized that dysregulation of TGF-β1 expression and the downstream pathway might cause abnormalities in extravillous trophoblasts and lead to PAS.

TGF-β1 is thought to play an important role in cellular tissue invasion and vascular remodeling, and various mechanisms have been elucidated in several cancers [20,21]. TGF-β-1 is the most abundant isoform and is secreted by different cell types, such as alveolar macrophages and epithelial cells, neutrophils, fibroblasts, and endothelial cells. The canonical TGF-β1 signaling pathway is initiated by the active ligand binding to the TGF-β receptor II (TβRII), leading to the formation of a heteromeric complex of TGF-β receptor I (TβRI) and TβRII on the cell membrane. Both TβRI and TβRII exhibit serine/threonine kinase activity. The Smad family of proteins serves as the essential mediators of signaling downstream of the TGF-β1 receptor kinases. The binding of TGF-β to specific receptors on the cell surface activates Smad1, -2, -3, -5, and -9, via carboxy-terminal phosphorylation of each receptor. Smad2 and -3 act specifically downstream of only TGF-β1, and bone morphogenetic proteins (BMPs) form a complex with type II receptors and phosphorylate Smad1, -5, and -9. These Smad proteins are important transcriptional regulators for intracellular TGF-β family signaling [22,23,24]. Recently, a negative feedback loop linking Smad2/3 activation and Smad1/5/9 inhibition was identified [25]. Phosphorylated Smad2/3 and Smad1/5 have opposing functions [26]. These results imply that some complicated protein–protein interactions occur between Smad proteins. TGF-β1 is regarded as a possible brake on cell proliferation in the context of carcinogenesis. Our previous study found that blocking ubiquitin carboxyl-terminal hydrolase-L5 (UCHL5) activity using a deubiquitinating enzyme (DUB) inhibitor, bAP15, inhibited the phosphorylation of Smad2/Smad3 and TGF-β1 signaling in a concentration-dependent manner and induced apoptosis in ovarian cancer cells [27]. The TGF-β pathway is also thought to regulate the MAPK pathway, including ERK signaling. Angiogenesis and vascularization in the placenta may be closely regulated [28]. Therefore, we hypothesized that abnormal TGF-β signaling in extravillous trophoblasts, which constitute the placenta, plays a major role in the pathogenesis of PAS. Abnormalities in invasive capacity and angiogenesis may be caused by Smad protein interactions. In this preliminary study, we investigated the molecular mechanism of PAS and determined the effects of UCHL5 on the invasive and angiogenic potential of extravillous trophoblasts.

## 2. Results

### 2.1. UCHL5 Regulates the TGF-β1-Smad2 Pathway and the Invasive Potential of Extravillous Trophoblast Cells

The TGF-β1 signaling pathway involves Smad-dependent signaling and several non-canonical pathways that potentially have different effects on the pathogenesis of PAS. To investigate its mechanism of action in the placenta, we investigated the role of UCHL5 in Smad2 phosphorylation in extravillous trophoblasts. We used two trophoblast cell lines—HTR-8/Svneo, derived from immortalized first-trimester placental villi, and JEG-3, differentiated from human choriocarcinoma cells. The same experiments were performed with both cell lines to test our hypothesis. We previously reported that UCHL5, a DUB, regulates Smad2 ubiquitination and induces Smad2 phosphorylation [27]. We investigated whether a similar mechanism occurs in extravillous trophoblasts and found that bAP15, an inhibitor of UCHL5, negatively regulates the phosphorylation of Smad2 in a concentration-dependent manner. Adding TGF-β1 (5 ng/mL) further activated Smad2, which was negatively regulated by b-AP15 (Figure 1).

Next, the effect of UCHL5 inhibition by bAP15 in each cell line was examined against changes in Smad2 expression by immunofluorescence staining. As shown in Figure 2, Smad2 was expressed in the nuclei of both JEG-3 and HTR-8/SVneo cells, and phosphorylated Smad2 (pSmad2) tended to be highly expressed in the nuclei, although there may have been some cell-specific differences after stimulation with TGF-β1.

We then analyzed the effect of UCHL5 inhibition by bAP15 treatment on the invasive potential of the two trophoblast cell lines. Cell invasiveness and migration are associated with pathophysiological processes, such as cancer metastasis. These processes include changes in cell structure and cytoskeletal dynamics, expression of adhesion molecules, and activation of EMT signaling. As shown in Figure 3, in line with our previous results in ovarian cancer cell lines, the addition of bAP15 inhibited UCHL5 and reduced cell invasiveness, as determined by Matrigel cell invasion experiments. Compared to JEG-3 cells, HTR-8/SVneo cells showed reduced invasion capacity; however, bAP15 treatment decreased the invasiveness of both cell lines.

### 2.2. Inhibition of UCHL5 Activates Smad1/5/9 and the ERK Pathway, a Non-Canonical Pathway for TGF-β Signaling

The TGF-β and BMP pathways signal downstream from cell surface receptors to the Smad proteins, and Smad2 or Smad1/5/9 could potentially function as downstream elements in both TGF-β and BMP pathway. Therefore, we examined whether inhibiting the TGF-β-Smad2 pathway using UCHL5 would alter the expression levels of Smad1/5/9 in JEG-3 and HTR-8/SVneo cell lines. JEG-3 cells exhibited low expression levels of phosphorylated Smad1/5/9 in the unstimulated state but showed increased expression under TGF-β stimulation (Figure 4A). In contrast, phosphorylated Smad1/5/9 accumulation was observed in HTR-8/SVneo cells, and phosphorylated Smad1/5/9 was activated after UCHL5 inhibition. No significant change was detected in the total amount of phosphorylated Smad1/5/9 proteins after UCHL5 inhibition and TGF-β1 stimulation in HTR-8/SVneo cells by Western blotting. Interestingly, UCHL5 inhibition significantly upregulated the activated phosphorylated Smad1/5/9 proteins in the plasma membrane of JEG-3 cells (Figure 4C). A further increase in phosphorylated Smad1/5/9 accumulation was observed in the nucleoli after UCHL5 inhibition under TGF-β stimulation (Figure 4C,D). Similarly, in HTR-8/SVneo cells, increased accumulation of phosphorylated Smad1/5/9 was observed in the nucleolus and the plasma membrane (Figure 4D). Our findings suggest that Smad1/5/9 could potentially bind to TGF-β receptors.

Next, we focused on Smad1, which is thought to promote the invasive and metastatic potential of certain types of cancer, and examined the expression level of Smad1 in JEG-3 and HTR-8/SVneo cells under UCHL5 inhibition. Both phosphorylated Smad1 and Smad1 expression levels in these cells decreased with increasing concentrations of bAP15 in the presence of TGF-β1 (Figure 4E,F). Specifically, the inhibition of UCHL5 resulted in a significant decrease in the expression level of Smad1 after TGF-β1 stimulation in JEG-3 and HTR-8/SVneo cells (Figure 4G,H).

In JEG-3 cells, the addition of TGF-β significantly increased the invasive potential of the cells (Figure 3A), whereas in HTR-8/SVneo cells, the effect of TGF-β on cell invasiveness was moderate. Therefore, we examined the effect of UCHL5 inhibition with or without TGF-β stimulation on ERK phosphorylation, which plays a conserved role in promoting cell migration and invasion in JEG-3 and HTR-8/SVneo cells. Contrary to our expectations, UCHL5 inhibition with bAP15 activated ERK phosphorylation in both cell lines (Figure 5A,B). Interestingly, this trend was more pronounced in HTR-8/SVneo cells than in JEG-3 cells (Figure 5B).

Phosphorylated ERK was upregulated in a bAP15 concentration-dependent manner, and the phosphorylated ERK(pERK)/ERK ratio and the phosphorylated Smad1/Smad1 ratio were increased by UCHL5 inhibition.

### 2.3. UCHL5 Inhibition Induces Extravillous Trophoblasts to Produce the Vascular Endothelial Growth Factor (VEGF) and Regulate Vascular Remodeling

As the remodeling of maternal uterine spiral arteries by invading the extravillous trophoblast is crucial for adequate blood supply to the fetus, we sought to validate the effect of bAP15 treatment on vascular remodeling. We treated the trophoblast lines with 5 μM bAP15 for 24 h. After the treatment, the VEGF concentration in the culture supernatant was measured using an enzyme-linked immunosorbent assay. Treatment with bAP15 increased VEGF concentrations in the culture supernatant, albeit with some variation, in both JEG-3 (Figure 6A) and HTR-8/SVneo cells (Figure 6B).

Since bAP15 treatment inhibited UCHL5 expression, negatively regulated Smad2 expression, and activated the ERK signaling pathway, we further determined the angiogenic potential of the supernatant using human umbilical vein endothelial cells (HUVEC).

Interestingly, HUVECs treated with fresh trophoblast culture medium containing 5 μM bAP15 showed clumping and probably underwent apoptosis (Figure 6C); however, incubating the culture media containing bAP15 for 24 h at 37 °C induced angiogenesis and vascular formation in HUVECs, suggesting that the effect of bAP15 disappeared after 24 h (Figure 6D). JEG-3 and HTR-8/SVneo cell culture supernatants containing bAP15 induced angiogenesis in HUVECs; however, vascular formation was insufficient, and small vascular aggregates were observed (Figure 6D,E). VEGF in the culture supernatant may have had an effect, but it was likely not sufficient to induce vascular tube formation.

## 3. Discussion

The aim of this study was to investigate the potential role of TGF-β and phosphorylated Smad proteins in the pathogenesis of PAS. We used two gestational trophoblast-derived cell lines to investigate the pathogenesis of the placenta. We found that the inhibition of UCHL5, a Smad2 deubiquitinating enzyme, negatively affected the TGF-β-Smad2 pathway. In JEG-3 cells, Smad2 was expressed mainly in the cytoplasm but was shifted to the nucleus when stimulated with TGF-β, as previously reported [27]. Furthermore, the nuclear expression of Smad2 decreased when the cells were treated with bAP15, as in our previous study of ovarian cancer cell lines [27,29]. A similar trend was observed in HTR-8/SVneo cells. Additionally, Smad2 was critical for cell invasiveness in the JEG-3 and HTR-8SVneo cells, in line with previous results [30,31,32,33,34,35,36,37]. In contrast to Smad2 expression, Smad1/5/9, downstream of the BMP pathway, showed increased expression upon UCHL5 inhibition, and phosphorylated Smad1/5/9 was significantly upregulated in the plasma membrane, but not in the nucleus, upon bAP15 treatment, and in the nucleoli under TGF-β stimulation. We also found that bAP15 has been reported to inhibit not only UCHL5 but also deubiquitinating enzymes such as USP14, although the effect may be weak [27]. Although we could not examine this in the present study, deubiquitinating enzymes such as UCHL5 and USP14 may be involved in Smad1/5/9 signaling. Nevertheless, we discovered that bAP15 has an effect on Smad1/5/9. In osteoblasts, activin A induces differentiation from preosteoblasts through the ALK1-Smad1/5/9 pathway downstream of the TGF-β pathway [38]. However, the role of this pathway in placentation has not been delineated.

Smad1/5/9 forms complexes with cell surface type II receptors [24,39], and TGF-β1 stimulation may induce complex formation, as observed in this study. Smad2 and Smad1/5/9 are called receptor-regulated Smads (R-Smads), and ligand-induced phosphorylation of these Smad proteins is essential to receptor Ser/Thr kinase-mediated TGF-β signaling [40]. In our study, phosphorylated Smad1 was highly expressed when cells were treated with low concentrations of bAP15 (1 μM) when Smad2 was not downregulated. However, at concentrations of bAP15 (5 μM) that inactivated Smad2, expression of Smad1 was not observed. Smad1, -5, and -9 functions are not yet known; however, the effects of UCHL5 inhibition on each of them likely differ.

In preeclampsia, which is a non-PAS-like disorder but a typical abnormal placental disorder, the abnormal expression of Smad2 affects extrachorionic villus differentiation, which in turn affects soluble Flt-1 activity. PAS is a condition of placental adhesion. Although pathologically dissimilar, the two conditions share a common abnormality in chorionic villi cells [30,41]. Under hypoxic conditions, large amounts of sFlt-1, which inhibits angiogenesis, and other factors are secreted into the intervillous space, inhibiting maternal neovascularization and causing vascular endothelial damage. An association between soluble Flt-1 and TGF-β1/Nox/p38 has also been shown [42]. Surgery-related oxidative stress profoundly affects normal cells [43], and cesarean-section wounds in the uterus may also be affected locally [44]. Although we did not use an experimental system to induce oxidative stress in this study, we examined the protein expression and signaling pathways induced by stimulation of the TGF-β pathway and showed that the pathway is also involved in the invasiveness of trophoblast cells.

In this study, we also examined the angiogenic potential of the trophoblast culture supernatant under TGF-β stimulation. The angiogenic potential of the TGF-β signaling pathway has been demonstrated previously [45]. We saw that, unexpectedly, TGF-β stimulation increased VEGF expression. This result was contrary to our expectations since the addition of bAP15 negatively regulated Smad2 in the TGF-β pathway, but VEGF production is positively regulated by ERK activity through the ERK/ERG/p300 transcriptional network, and VEGF production in trophoblast cells may be dependent on the ERK signaling pathway [46]. ERK can be inhibited by Smad1/5 in stem cells [39], indicating that they may be related, albeit in different cell types. In PAS, it is known that, in addition to the invasive capacity of cells, abnormalities in the vascular architecture of the placenta also occur, and the vascular architecture is commonly examined by ultrasound and MRI [47]. Interestingly, we showed that although UCHL5 inhibition increased VEGF levels, it did not stimulate vascular endothelial cells to form vessels, indicating an essential role for UCHL5 in the formation of normal placenta.

The current study has certain limitations. First, we were not able to examine the relationship between Smad protein expression levels and TGF-β1 expression in clinical specimens of PAS. Further, we were not able to examine invasive and angiogenic potentials in experimental animal models, such as mouse models. However, our findings provide the basis for future research.

In summary, the TGF-β-UCHL5-Smad2 pathway is essential for the invasion of extravillous trophoblasts, which play a vital role in the formation of the placenta, from the implantation site toward the maternal endometrium to the smooth muscle and endothelium of the uterine spiral artery vasculature, where Smad1/5/9 acts as regulatory factors. The TGF-β-UCHL5-Smad2 pathway is also regulated by the ERK signaling pathway, which promotes angiogenesis and vascularization. The disruption of these signaling pathways in placenta accreta may result in abnormal placental invasion and angiogenesis. Although clinical specimens were not used, we successfully identified common signaling pathways using two trophoblast cell lines, JEG-3 and HTR-8/SVNeo. There is currently no cure for PAS, and the underlying mechanisms remain unknown. Identifying the detailed molecular mechanisms underlying PAS pathogenesis is critical for developing effective treatments.

## 4. Materials and Methods

### 4.1. Antibodies and Reagents

TGF-β1 (209-16554) was purchased from Fujifilm Wako Pure Chemical Industries (Osaka, Japan), and bAP15 (11324) was purchased from Cayman Chemical (Ann Arbor, MI, USA). The following antibodies were used: anti-β-actin (sc-47778), anti-UCHL5 (sc-271002), anti-Vimentin (sc-6260; Santa Cruz Biotechnology, Dallas, TX, USA), anti-E-cadherin (610181; BD Bioscience, San Jose, CA, USA), anti-smad1 (6944), anti-Phospho-Smad1 (5753), anti-Smad2 (5339), anti-Phospho-Smad2 (3108), anti-Phospho-Smad2 (18338), anti-Smad3 (9523), anti-Phospho-Smad3 (9520), anti-Smad4 (38454), anti-smad5 (12534), anti-Phospho-smad1/5 (9516), anti-Phospho-smad1/5/9 (13820), anti-ERK (9102), anti-Phospho-ERK (9101), anti-AKT (4691), anti-Phospho-AKT (4060; Cell Signaling Technology, Danvers, MA, USA), and anti-smad7 (216428; Abcam Inc., Toronto, ON, Canada).

### 4.2. Cell Culture

Gestational trophoblast-derived cells were used. The gestational choriocarcinoma cell line JEG-3 and trophoblast cell line HTR-8/SVneo were obtained from the American Type Culture Collection (Manassas, VA, USA). JEG-3 cells were cultured in EMEM (Fujifilm Wako Pure Chemical Industries, Osaka, Japan) supplemented with 10% fetal bovine serum (Gibco), and HTR-8/SVneo cells were cultured in RPMI (Gibco, Grand Island, NY, USA) supplemented with 10% fetal bovine serum (Gibco). The cells were maintained at 37 °C in a 5% CO_2_ incubator.

### 4.3. Western Blotting

Cells (4 × 10^6^) were pretreated with or without bAP15 (5 μM) for 1 h and then with TGF-β1 (0 or 5 ng/mL) for 2 h. Equal amounts of proteins were fractionated by sodium dodecyl sulfate-polyacrylamide gel electrophoresis and transferred onto a polyvinylidene difluoride membrane (Millipore, Bedford, MA, USA). The membranes were blocked and incubated with the primary antibodies, followed by the appropriate secondary antibodies. For all analyses, we regenerated the membrane by stripping off the antibodies and reprobing to assure the accuracy of each result. The signal was detected using an ImageQuant LAS 4000 Mini instrument (GE Healthcare, Wauwatosa, WI, USA). Protein expression levels were quantified for each band using an ImageQuant LAS 4000 Mini instrument, and the relative quantity of protein with respect to actin (Figure 1 and Figure 4) or ERK (Figure 5) was calculated.

### 4.4. Immunofluorescence

JEG-3 or HTR-8/SVneo cells (1 × 10^5^) were treated with bAP15 and TGF-β1, as described above. After treatment, cells were fixed in 4% paraformaldehyde for 30 min and blocked with 6% bovine serum albumin (Gibco, Grand Island, NY, USA) for 1 h. Cells were immunostained with a primary antibody for 1 h at room temperature and incubated with the corresponding fluorescent probe-conjugated secondary antibody for 1 h in the dark. The secondary antibodies used were Alexa Fluor 488 (150105; Abcam Inc., Toronto, ON, Canada) for Smad2 and Smad3 and Alexa Fluor 594 (150076; Abcam Inc.) for p-smad2 and p-smad1/5/9. Images were captured using a confocal fluorescence microscope (FV10i; Olympus, Tokyo, Japan).

### 4.5. Endothelial-like Network Formation Assay

JEG-3 or HTR-8/SVneo cells (2 × 10^5^) were incubated for 24 h after treatment with bAP15 and TGF-β1. Chilled Matrigel solution (Corning, Manassas, VA, USA; 500 μL/well) without air bubbles was added to 96-well plates and incubated at 37 °C for 1 h to solidify.

HUVECs (JCRB1458; 2 × 10^5^ cells/500 μL) were seeded in the cell culture supernatant (cells were treated as mentioned above), and a 24 h time-lapse video was recorded using a CellVoyager CQ1 confocal image cytometer (Yokogawa Electric Corp., Tokyo, Japan). The angiogenic properties were evaluated by measuring the number of tubular structures from vascular endothelial cells.

### 4.6. Measurement of VEGF

JEG-3 or HTR-8/SVneo cells (3 × 10^5^) were incubated for 24 h after treatment with bAP15 and TGF-β1. The cell culture supernatants were collected after 24 h. The supernatant was stored at −80 °C until use. The Human VEGF Quantikine ELISA Kit (R&D Systems, Minneapolis, MN, USA) was used to measure VEGF levels in the supernatant following the manufacturer’s instructions.

### 4.7. Matrigel Invasion Assay

A Matrigel invasion assay was performed following the manufacturer’s instructions: 0.5 mL of cell suspension (JEG-3: 3 × 10^5^ cells, HRT-8/SVneo: 1 × 10^5^ cells) in serum-free medium was seeded into the top of a Matrigel Invasion Chamber (354481; Corning). TGF-β and bAP15 were added to the top chamber. In the bottom chamber, we added EMEM (Fujifilm Wako Pure Chemical Industries, Osaka, Japan) for JEG-3 and RPMI (Gibco, Grand Island, NY, USA) for HRT-8/SVneo supplemented with 10% fetal bovine serum (Gibco). After incubation at 37 °C for 24 h, non-infiltrating cells were removed from the upper chamber, and cells attached to the lower chamber were stained using Diff-Quick reagent (Sysmex, Kobe, Japan).

### 4.8. Statistical Analysis

Statistical significances were determined in triplicate experiments for the control group and the UCHL5-inhibited group after the addition of bAP15 to evaluate and eliminate the possibility of heterogeneous variance, using the Kruskal–Wallis test in the GraphPad Prism 6 software (GraphPad, San Diego, CA, USA) and JMP 17 (SAS Institute, Tokyo, Japan) in independent experiments, with *p* < 0.05 considered significant.

## 5. Conclusions

In this study, we investigated the effect of inhibiting UCHL5, which regulates the TGF-β-Smad2 pathway, using bAP15, on the invasive ability and angiogenesis potential of trophoblasts. The results revealed the involvement of Smad1/5/9 downstream in the TGF-β pathway and the activation of the ERK pathway under TGF-β stimulation, suggesting a mechanism underlying PAS development and providing insights into the development of molecularly targeted therapeutics.

## Figures and Tables

**Figure 1 ijms-24-13706-f001:**
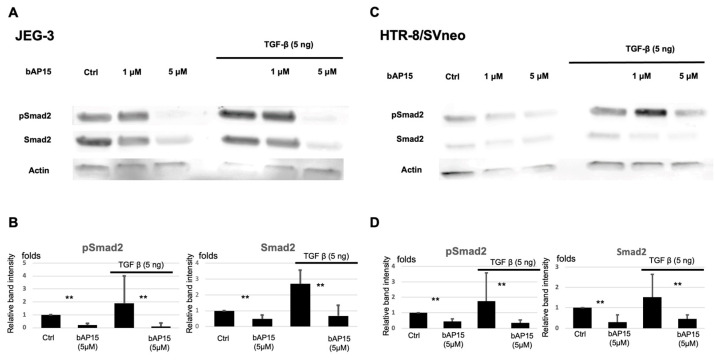
Ubiquitin carboxyl-terminal hydrolase-L5 (UCHL5) regulates the transforming growth factor (TGF-β)-Smad2 pathway in trophoblasts. JEG-3 (**A**,**B**) and HTR-8/SVneo (**C**,**D**) cells were pretreated with bAP15 (1 or 5 μM), a deubiquitinating enzyme (DUB) inhibitor, for 1 h, followed by TGF-β1 (5 ng/mL) for 2 h. Western blotting was performed to observe the expression levels of proteins related to the TGF-β pathway. The plots in (**B**,**D**) show the quantification of the blots in (**A**,**C**), respectively. Data are presented as mean ± SD from triplicate experiments. ** *p* < 0.05 versus control.

**Figure 2 ijms-24-13706-f002:**
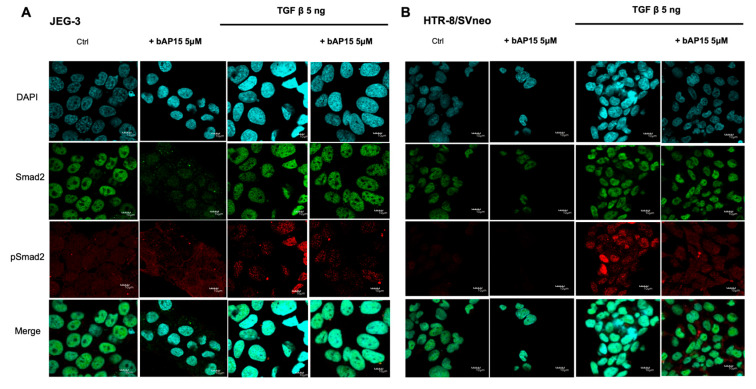
Ubiquitin carboxyl-terminal hydrolase-L5 (UCHL5) inhibition decreased Smad2 expression and phosphorylation. Immunofluorescence staining for Smad2 (green) and phosphorylated Smad2 (red) in JEG-3 (**A**) and HTR-8/Svneo (**B**) cells. Smad2 and phosphorylated Smad2 decreased in cells treated with bAP15 (5 μM) and stimulated with transforming growth factor (TGF)-β1 (5 ng/mL). Scale bar = 10 μm.

**Figure 3 ijms-24-13706-f003:**
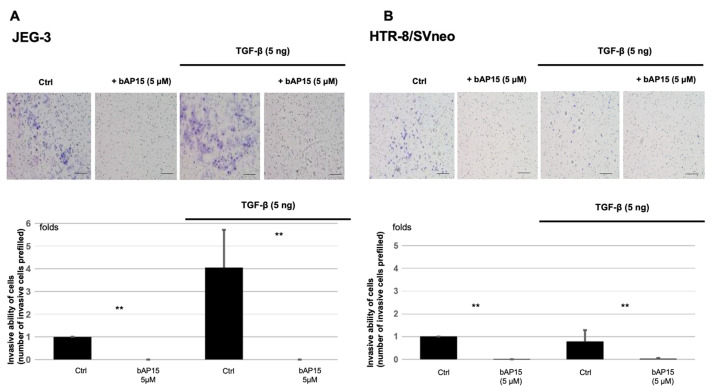
Transforming growth factor (TGF)-β-Smad2 pathway inhibition resulted in dysfunctional trophoblast invasiveness. Matrigel transwell invasion assays were performed to evaluate the effect of TGF β-Ubiquitin carboxyl-terminal hydrolase-L5 (UCHL5)-Smad2 on cell invasiveness. Representative images of transwell membranes of JEG-3 (**A**) and HTR-8/Svneo (**B**) cells treated with or without bAP15 (5 μM), TGF-β1 (5 ng/mL), or bAP15 (5 μM) +TGF-β1 (5 ng/mL), for 24 h. The bottom panel shows the quantification of the number of invasive cells. Data are presented as mean ± SD from triplicate experiments. ** *p* < 0.05 versus control.

**Figure 4 ijms-24-13706-f004:**
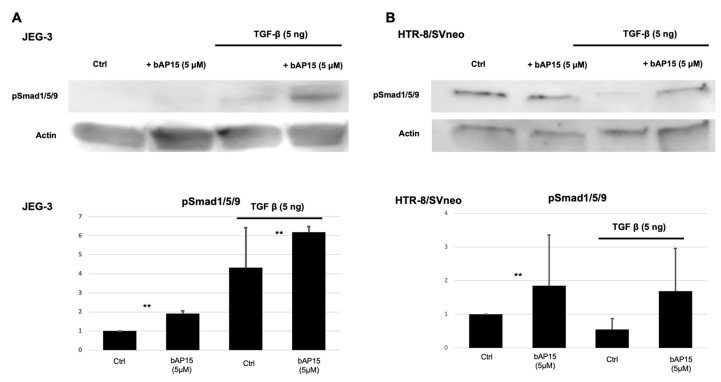
Inhibition of ubiquitin carboxyl-terminal hydrolase-L5 (UCHL5) regulates Smad1/5/9 phosphorylation. (**A**,**B**) Representative Western blot of membrane-associated proteins and (**C**,**D**) immunocytochemistry staining of phosphorylated Smad1/5/9 in JEG-3 and HTR-8/Svneo cells treated with or without bAP15 (5 μM), transforming growth factor (TGF)-β1 (5 ng/mL) stimulation following bAP15 treatment increased Smad1/5/9 phosphorylation, as determined via Western blotting and immunostaining in JEG-3 (**A**) and HTR-8/Svneo (**B**) cells. Since phosphorylated Smad1/5/9 was localized to the plasma membrane, the membrane-associated proteins were extracted, and the increased Smad1/5/9 phosphorylation was confirmed by Western blotting. To investigate the function of UCHL-5 in the TGF-β-induced signaling in JEG-3 and HTR-8/Svneo cells, they were pretreated with bAP15 (1 or 5 μM) for 1 h and then with TGF-β1 (5 ng/mL) for 2 h. Western blotting was performed to examine changes in Smad1 activation and expression levels. (**E**,**F**) Representative Western blots and (**G**,**H**) respective quantification plots of phosphorylated Smad1 in JEG-3 and HTR-8/Svneo cells pretreated with bAP15 (1 or 5 μM) for 1 h and then with TGF-β1 (5 ng/mL) for 2 h. Data are presented as the mean ± SD from triplicate experiments. ** *p* < 0.05, versus control.

**Figure 5 ijms-24-13706-f005:**
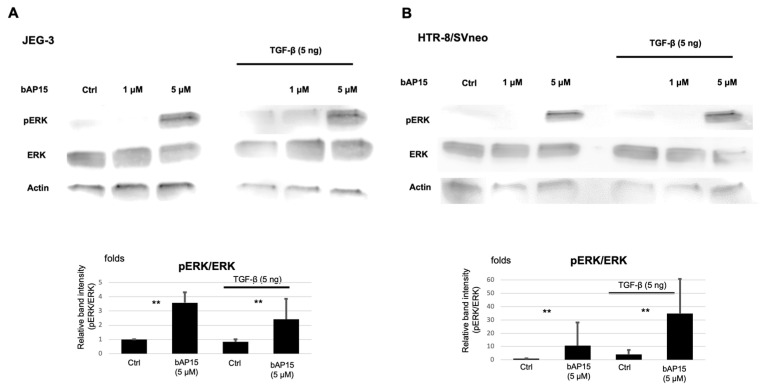
Ubiquitin carboxyl-terminal hydrolase-L5 (UCHL5) inhibition activates the ERK pathway in trophoblasts. Representative Western blots showing ERK expression and phosphorylation in JEG-3 (**A**) and HTR-8/Svneo (**B**) cells pretreated with bAP15 (1 or 5 μM) for 1 h and then with transforming growth factor (TGF)-β1 (5 ng/mL) for 2 h. Data are presented as the mean ± SEM/SD from triplicate experiments. ** *p* < 0.05 versus control.

**Figure 6 ijms-24-13706-f006:**
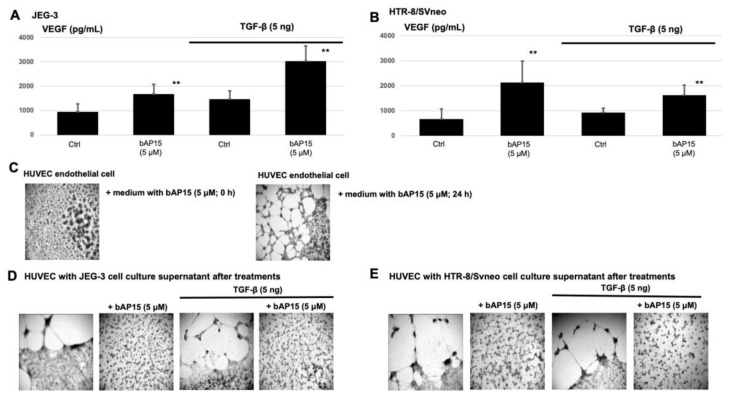
The transforming growth factor (TGF) β-ubiquitin carboxyl-terminal hydrolase-L5 (UCHL5)-Smad2 pathway modulates trophoblast angiogenesis. Vascular endothelial growth factor (VEGF) levels (pg/mL) in JEG-3 (**A**) and HTR-8/SVneo (**B**) cells treated with or without bAP15 (5 μM), TGF-β1 (5 ng/mL), or b-AP15 (5 μM) + TGF-β1 (5 ng/mL) for 24 h, determined using ELISA. (**C**) Representative micrograph showing human umbilical vein endothelial cells (HUVEC) treated with fresh trophoblast media + bAP15 (5 μM) or trophoblast media + bAP15 incubated for 24 h at 37 °C. Representative micrographs of HUVEC treated with cell culture supernatants with or without bAP15 (5 μM), TGF-β1 (5 ng/mL), or bAP15 (5 μM) + TGF-β1 (5 ng/mL) from JEG-3 (**D**) and HTR-8/Svneo (**E**) cells. Notably, under UCHL5 inhibition, the cells formed a shorter full-length network than the controls and formed some small vessels that lost orientation without building the reticular structure seen in controls. Data are presented as mean ± SD. ** *p* < 0.05 versus control.

## Data Availability

Not applicable.

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
