# Peer review of "The TGF-β/UCHL5/Smad2 Axis Contributes to the Pathogenesis of Placenta Accreta"

_ijms, 2023, doi:10.3390/ijms241813706_

Round 1

Reviewer 1 Report

Overall, the present study clarifies the molecular pathogenesis of an important topic in human obstetrics, through identifying the role of transforming growth factor (TGF)-β1-Smad pathway and investigating the intrinsic  relationship between the time- and dose-dependent inhibition of the ubiquitinating enzyme UCHL5  using a deubiquitinase inhibitor (bAP15) after TGF-β1 stimulation and the invasive and angiogenic  potential of two cell lines derived from extravillous trophoblasts, JEG-3 and HTR-8/SVneo. This study is interesting and provides useful information. However, there are some issues in manuscript organization, conclusion and some minor corrections in the context.

Specific comments

1.      Abstract

Line 15: please put bAP15 between practice.

Line 16: please identify JEG-3 and HTR-8/SVneo.

Line 18: please identify ERK.

Line 27: please rearrange the keywords alphabetically.

2.       Introduction

The hypothesis is clear. Introduction is comprehensible.

Line 40: replace “indicate” with “have indicated”

3.      Results

-      The position of results section just after introduction section is not familiar. Many sentences in the results section belong to the materials and methods section and this is not appropriate.

-      Results section should be mentioned in the past tense not the present tense. Please adjust.

-      Line 139-140: the authors should mention the results of the present study regardless of other studies, even if their past results.

4- Discussion

Well written and cover all the required points

5- Materials and methods

Line 352: which type of cells?

-Statistical analysis: statistical tests used in discrimination the significance difference should be described comprehensively.

6- Conclusion

Line 408-411: this data is already available, and the authors did not examine it; therefore, it should be omitted.

Conclusion should cover only what the authors obtained from their trials. I suggest reformulation of the conclusion based on what I mentioned above.

The quality of English is acceptable.

Author Response

Overall, the present study clarifies the molecular pathogenesis of an important topic in human obstetrics, through identifying the role of transforming growth factor (TGF)-β1-Smad pathway and investigating the intrinsic  relationship between the time- and dose-dependent inhibition of the ubiquitinating enzyme UCHL5  using a deubiquitinase inhibitor (bAP15) after TGF-β1 stimulation and the invasive and angiogenic  potential of two cell lines derived from extravillous trophoblasts, JEG-3 and HTR-8/SVneo. This study is interesting and provides useful information. However, there are some issues in manuscript organization, conclusion and some minor corrections in the context.

Reply: We thank the reviewer for commending our work and for the constructive comments that have helped to substantially improve the quality of our manuscript.

Specific comments

  1. Abstract

Line 15: please put bAP15 between practice.

Line 16: please identify JEG-3 and HTR-8/SVneo.

Line 18: please identify ERK.

Line 27: please rearrange the keywords alphabetically.

Reply: We have modified the text accordingly.

  1. Introduction

The hypothesis is clear. Introduction is comprehensible.

Line 40: replace “indicate” with “have indicated”

Reply: We have modified the text accordingly.

  1. Results

-      The position of results section just after introduction section is not familiar. Many sentences in the results section belong to the materials and methods section and this is not appropriate.

-      Results section should be mentioned in the past tense not the present tense. Please adjust.

Reply: The sections are ordered according to the journal guidelines. We have, however, modified the text for clarity and to account for the reviewer’s comment.

-      Line 139-140: the authors should mention the results of the present study regardless of other studies, even if their past results.

Reply: We have revised the text according to the reviewer’s comment.

4- Discussion

Well written and cover all the required points

Reply: We thank the reviewer for the positive feedback.

5- Materials and methods

Line 352: which type of cells?

Reply: We have added this description for clarity: line 346-348.

We used two gestational trophoblast-derived cell lines to investigate the pathogenesis of the placenta.

-Statistical analysis: statistical tests used in discrimination the significance difference should be described comprehensively.

Reply: We have added more details on the statistics used, as per the reviewer’s comment in line 408-412.

Statistic significances were determined in triplicate experiments for the control group and the UCHL5 inhibited group after the addition of bAP15 to evaluate and eliminate the possibility of heterogeneous variance, using the Kruskal-Wallis test in the GraphPad Prism 6 software (GraphPad, San Diego, CA) and JMP 17 (SAS Institute, SAS Institute, Japan) in independent experiments. Tokyo, Japan), with p < 0.05 considered significant.

6- Conclusion

Line 408-411: this data is already available, and the authors did not examine it; therefore, it should be omitted.

Conclusion should cover only what the authors obtained from their trials. I suggest reformulation of the conclusion based on what I mentioned above.

Reply: We have reformulated the conclusions based on the reviewer’s comment.

Reviewer 2 Report

The Authors investigated the TGF-beta/Smad pathway in the placenta accreta spectrum.

The article is interesting and quite well-written.

The Discussion and Materials & Methods Sections are not so well separated. Please try to rearrange these two sections, clearly recognizing how you did what you did and what results you obtained. In particular, no mention is made of why the Authors chose those specific cell lines to investigate PAS. Please, disclose the reason for that choice. I agree with the Authors that a histological specimen (not "clinical," as the Authors wrote - line 325) would be a better substrate to test their hypothesis.

I found some statements that are somehow out of context or not adequately explained, i.e., preeclampsia is not at all similar to PAS: as a matter of fact, one the complication of preeclampsia is abruptio placenta (the placenta spontaneously detaching from the decidual substrate and the uterus wall before the due time, due to superficial implantation), while in PAS placenta is so deeply "attached" to the uterine wall that must be removed manually. I suggest checking the literature more accurately or arguing better eventual similarities. 

Author Response

Comments and Suggestions for Authors

The Authors investigated the TGF-beta/Smad pathway in the placenta accreta spectrum.

The article is interesting and quite well-written.

The Discussion and Materials & Methods Sections are not so well separated. Please try to rearrange these two sections, clearly recognizing how you did what you did and what results you obtained. In particular, no mention is made of why the Authors chose those specific cell lines to investigate PAS. Please, disclose the reason for that choice. I agree with the Authors that a histological specimen (not "clinical," as the Authors wrote - line 325) would be a better substrate to test their hypothesis.

Reply: We thank the reviewer very much for the valuable comments. We have further performed various revisions in the Discussion and Materials and methods sections as per reviewers’ suggestions; however, we have refrained from any major revisions to these sections as we believe that the current structuring serves the purpose of presenting our research findings effectively. If there are specific points within these sections that you find unclear or overlapping, we would be happy to address those concerns directly by making targeted revisions or by providing additional explanations where needed.

I found some statements that are somehow out of context or not adequately explained, i.e., preeclampsia is not at all similar to PAS: as a matter of fact, one the complication of preeclampsia is abruptio placenta (the placenta spontaneously detaching from the decidual substrate and the uterus wall before the due time, due to superficial implantation), while in PAS placenta is so deeply "attached" to the uterine wall that must be removed manually. I suggest checking the literature more accurately or arguing better eventual similarities. 

Reply: We thank the reviewer very much for the valuable comments. We have added the following description in line 287-291:

In preeclampsia, which is non-PAS-like disorder but a typical abnormal placental disorder, abnormal expression of Smad2 affects extrachorionic villus differentiation, which in turn affects soluble Flt-1 activity. PAS is a condition of placental adhesion. Although pathologically dissimilar, the two conditions share a common abnormality in chorionic villi cells [30,37].
